# PathwayLLM: Explainable Clinical Trajectory Modeling with Structured Pathways for Sepsis Prediction

**Zhengqiu Yu** [1 2]  **Yueping Ding** [3]  **Xiangrong Liu** [4 2 5]

## Abstract

Patient-level sepsis prediction requires models that track clinical deterioration over time and integrate heterogeneous structured evidence from electronic health records. We present PathwayLLM, a trajectory-based framework that grounds prediction on temporal signals, graph-structured evidence, and pathway-level clinical information derived from statistical dependency discovery. PathwayLLM follows a three-stage design. First, each observation window is encoded from multiple structured views, including physiological measurements, temporal dynamics, a heterogeneous patient–diagnosis–medication graph, and dependency-derived pathway signals. Second, these representations are injected into a pretrained language model as auxiliary contextual embeddings so that risk prediction and evidence-conditioned explanations can be learned jointly. Third, a Clinical Trajectory LSTM with Deterioration Attention aggregates window-level representations to highlight critical deterioration points and produce patient-level risk scores. On MIMIC-IV (15,410 ICU patients; 8.45% sepsis prevalence), PathwayLLM achieves AUROC 0.891 and AUPRC 0.724, outperforming strong time-series and pretrained baselines. External validation on eICU achieves AUROC 0.842 zero-shot and 0.867 after light fine-tuning. Ablation studies indicate that trajectory aggregation and structured clinical signals are key contributors, and clinician review suggests coherent, interpretable, and clinically relevant explanations.

## 1. Introduction

Sepsis, defined by the Sepsis-3 consensus as life-threatening organ dysfunction caused by dysregulated host response to infection (Singer et al., 2016), represents one of the most challenging clinical conditions in intensive care medicine. With an estimated 48.9 million cases and 11 million sepsis-related deaths globally each year (Rudd et al., 2020), sepsis accounts for approximately 20% of all deaths worldwide. Delays in antimicrobial therapy significantly increase mortality (Seymour et al., 2017), underscoring the clinical imperative for early prediction and timely intervention.

Despite advances in machine learning for clinical prediction (Fleuren et al., 2020; Shashikumar et al., 2021), sepsis presents unique challenges. Its complex, heterogeneous pathophysiology involves cascading interactions among infection, inflammation, coagulation, and organ dysfunction, demanding models that capture both population-level patterns and patient-specific temporal dynamics. Clinical decision-making benefits from mechanistic understanding: clinicians must infer active disease mechanisms, predict intervention effects, and identify plausible clinical relationships (Pearl, 2009; Schölkopf et al., 2021). Meanwhile, clinical adoption requires interpretability beyond simple feature attribution (Sadeghi et al., 2024). Physicians need explanations that connect clinical indicators to disease processes in ways that align with medical training and enable reasoning verification.

Current sepsis prediction approaches address these challenges only partially. Time-series models (LSTMs, GRUs, Transformers) capture temporal dependencies within individual trajectories but cannot incorporate population-level disease patterns. Graph neural networks encode patient-diagnosis-medication relationships through heterogeneous information networks and relational message passing (Schlichtkrull et al., 2018; Oss Boll et al., 2024), enabling similarity-based reasoning but learning correlational patterns without explicit structural interpretation. Recent advances in large language models offer opportunities to

[1]School of Medicine, Xiamen University, Xiamen 361005, PR China [2]National Institute for Data Science in Health and Medicine, Xiamen University, Xiamen 361005, PR China [3]Department of Critical Care Medicine, The Second Affiliated Hospital of Zhejiang Chinese Medical University, Hangzhou 310000, PR China [4]Department of Computer Science and Technology, Xiamen University, Xiamen 361005, PR China [5]Xiamen Key Laboratory of Intelligent Storage and Computing, Xiamen University, Xiamen 361005, PR China. Correspondence to: Xiangrong Liu <xrliu@xmu.edu.cn>.

*Proceedings of the 43rd International Conference on Machine Learning*, Seoul, South Korea. PMLR 306, 2026. Copyright 2026 by the author(s).

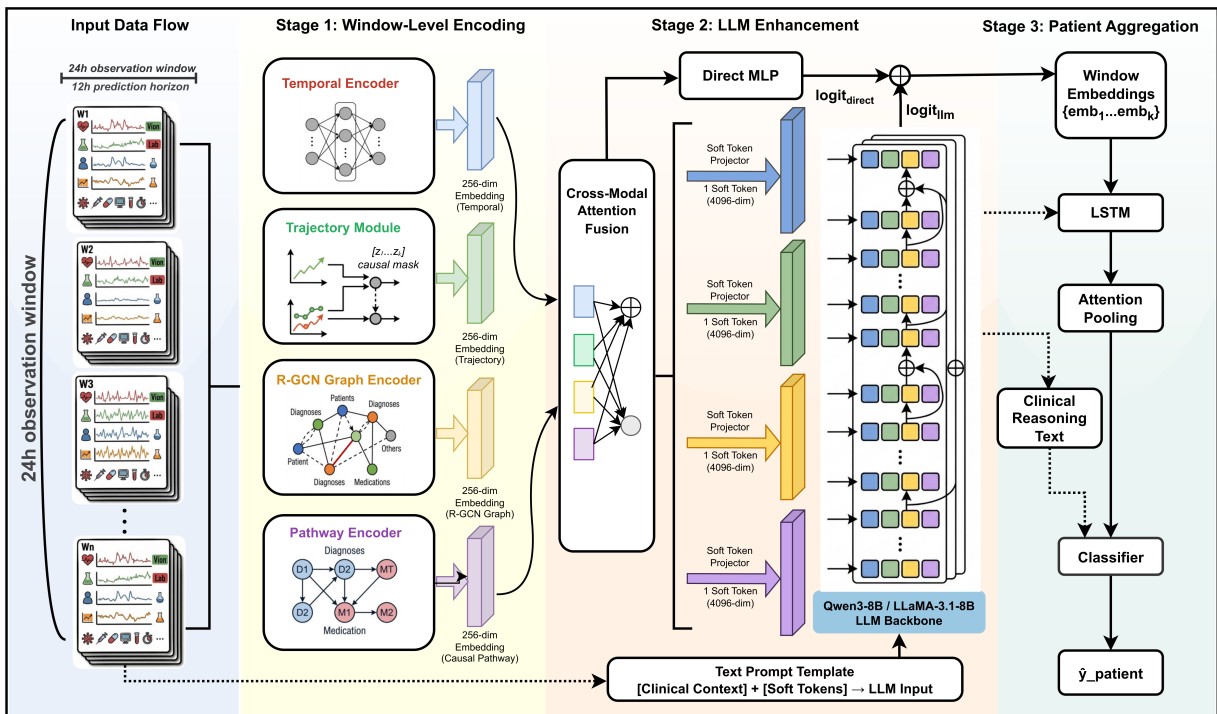

*Figure 1.* **PathwayLLM Three-Stage Architecture.** Stage 1: Four encoders (Temporal, Trajectory, R-GCN, Pathway) extract 256-dim embeddings per window. Stage 2: Cross-Modal Fusion produces fused representation for Direct Path and original embeddings for LLM Path with four modality-specific auxiliary embeddings (soft tokens); both paths merge via $\alpha_{\text{fuse}}$-weighted fusion. Stage 3: LSTM and attention pooling aggregate window embeddings into patient-level prediction.

address interpretability challenges through natural language explanations (Singhal et al., 2023; Nori et al., 2023), but direct application to structured EHR prediction remains difficult (Wornow et al., 2023) as LLMs struggle with high-dimensional temporal data.

We propose PathwayLLM for structured evidence-grounded patient-level sepsis prediction (Fig. 1). Our contributions include: (1) A multi-view encoding architecture combining temporal physiology, cross-window trajectory dynamics, heterogeneous patient-diagnosis-medication graphs, and pathway-level signals derived from dependency discovery. (2) A Clinical Trajectory LSTM with Deterioration Attention that aggregates window-level representations to identify critical deterioration points and produce patient-level risk scores. (3) Evidence-conditioned text explanations generated jointly with risk prediction, leveraging LLM reasoning capabilities. (4) Extensive experimental validation including quantitative evaluation and clinician assessment of explanation quality.[1]

---

[1]Code is available at: https://github.com/leanqon/PathwayLLM

## 2. Related Work

Sepsis prediction has evolved from rule-based clinical scoring systems to sophisticated deep learning architectures. Traditional clinical scores including SOFA and qSOFA (Singer et al., 2016) provide clinically interpretable risk assessments but achieve limited discriminative performance (AUROC 0.5–0.65). Machine learning approaches brought improvements through gradient boosting and random forests on engineered clinical features (Mao et al., 2018). Prospective deployment studies such as TREWS evaluated operational sepsis early-warning systems in multi-site clinical settings (Adams et al., 2022). Deep learning architectures for time-series clinical data include LSTMs, GRUs, temporal convolutional networks (Bai et al., 2018), and Transformers (Vaswani et al., 2017). EHR-specialized pre-trained models (BEHRT (Li et al., 2020), Med-BERT (Rasmy et al., 2021), CEHR-BERT (Pang et al., 2021)) leverage large-scale clinical data. The COMPOSER system (Shashikumar et al., 2021) demonstrated conformal prediction for early sepsis warning and abstention under unfamiliar cases.

## 2.1. Graph Neural Networks for EHR

Graph neural networks encode structured EHR relationships through heterogeneous information networks capturing patient-diagnosis-medication structures, with recent surveys summarizing their use in EHR-based clinical risk prediction (Oss Boll et al., 2024). R-GCN (Schlichtkrull et al., 2018) handles multiple edge types via relation-specific transformations. MulT-EHR (Chan et al., 2024) demonstrated effective outcome prediction from heterogeneous EHR graphs.

## 2.2. Dependency Discovery in Healthcare

Statistical dependency discovery provides frameworks for identifying structural relationships from observational data (Pearl, 2009). Constraint-based methods such as the PC algorithm (Spirtes et al., 2000) iteratively test conditional independence to identify graph structures. Recent advances have been applied to healthcare for identifying treatment effects and disease mechanisms (Prosperi et al., 2020). Our work applies these methods to diagnosis-medication relationships, extracting patient-specific pathway activation patterns that provide structured domain knowledge grounding model predictions.

## 2.3. Large Language Models for Clinical Prediction

Large language models show promise in medical applications (Singhal et al., 2023; Nori et al., 2023), but integrating LLMs with structured EHR data remains challenging (Wornow et al., 2023). NYUTron (Jiang et al., 2023) predicts outcomes from clinical notes, GRASP (Kirchler et al., 2026) improves structured-EHR transfer through LLM-derived semantic code embeddings, and KARE (Jiang et al., 2025) retrieves knowledge graph communities to support LLM reasoning for healthcare prediction. Our approach instead grounds a frozen LLM with patient-specific temporal, graph, and dependency-pathway evidence via soft tokens and LoRA adaptation (Hu et al., 2022).

# 3. Methods

## 3.1. Problem Definition

Given a patient's clinical trajectory spanning multiple 24-hour observation windows during their ICU stay, our framework operates at two complementary levels. At the window level, each observation window encodes clinical features for sepsis risk estimation. At the patient level, our model aggregates all window-level representations to predict whether the patient will develop sepsis during their ICU stay. Formally, let $\mathcal{W}_p = \{\mathbf{X}_1, \mathbf{X}_2, \ldots, \mathbf{X}_K\}$ denote the sequence of $K$ observation windows for patient $p$, where each $\mathbf{X}_k$ contains 199-dimensional clinical features. The task is to learn $f : \mathcal{W}_p \rightarrow [0, 1]$ estimating $P(\text{sepsis} \mid \mathcal{W}_p)$. We define $y_k^{(w)} = 1$ if sepsis onset occurs within 12 hours after window $k$ ends, and $y^{(p)} = 1$ if any window satisfies $y_k^{(w)} = 1$. This setting constitutes risk stratification within the suspected-infection population, matching clinical decision points after infection is suspected. Beyond prediction, we require interpretable clinical reasoning explaining predictions through activated dependency pathways and temporal attention weights.

## 3.2. Data and Cohort

We used MIMIC-IV v3.1 (Johnson et al., 2023), a comprehensive critical care database from Beth Israel Deaconess Medical Center. Inclusion criteria: (1) adults (age $\geq 18$); (2) ICU stay 48–168 hours; (3) first ICU stay only; (4) suspected infection (blood culture within $-24$ to $+72$h of antibiotics). This yielded 15,410 patients. Sepsis was defined per Sepsis-3 (Singer et al., 2016): suspected infection plus SOFA increase $\geq 2$ within $\pm 48$h. Among 15,410 patients, 1,302 (8.45%) developed sepsis. We constructed sliding windows (24h duration, 12h stride), generating 71,724 windows. Each window is labeled positive if sepsis onset occurs within 12 hours after the window ends, providing a clinically actionable prediction horizon. Critically, for sepsis-positive patients, we censor observation at onset time: only windows ending before onset are used for patient-level aggregation, preventing temporal leakage from post-onset observations. For negative patients, all windows are used. The patient-level prediction aggregates window-level representations through Stage 3's attention mechanism, which learns to weight windows indicative of imminent deterioration more heavily. Data were split at patient level (70/15/15%) using stratified sampling into training (10,787 patients, 8.45% sepsis), validation (2,311 patients, 8.44% sepsis), and test (2,312 patients, 8.48% sepsis) sets, maintaining consistent prevalence across splits.

Each window comprises 199 features: 128 structured clinical features (demographics, vital signs with 5 statistics, labs, treatment indicators), 51 diagnosis features (top 50 ICD-10 + count), and 20 medication features (19 ATC categories + count). To prevent leakage, only *admission diagnoses* are used. See Appendices A and B for cohort selection and feature details.

## 3.3. Knowledge Graph Construction

We construct two complementary graph structures from training data (Figure 2). The Heterogeneous Information Network (HIN) $\mathcal{G} = (\mathcal{V}, \mathcal{E})$ encodes patient-diagnosis-medication relationships: 10,787 patient nodes (training set), 50 diagnosis nodes, 19 medication nodes, with 139,398 edges. Patient similarity via meta-paths (Patient-Diagnosis-Patient and Patient-Medication-Patient) captures degree-

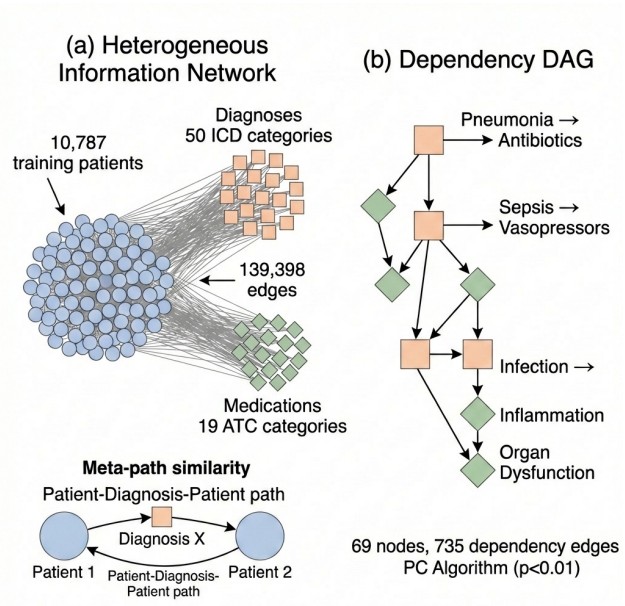

*Figure 2.* Knowledge Graph Structures. Left: HIN with 10,787 patients, 50 diagnoses, 19 medications, 139,398 edges. Right: Dependency DAG with 735 edges among clinical concepts (diagnoses, medications, physiological states) via PC algorithm.

normalized co-occurrence. A two-layer R-GCN with 30 basis decompositions computes embeddings as:

$$\mathbf{h}_v^{(l+1)} = \sigma\left(\sum_{r \in \mathcal{R}} \sum_{u \in \mathcal{N}_r(v)} \frac{1}{|\mathcal{N}_r(v)|} \mathbf{W}_r^{(l)} \mathbf{h}_u^{(l)} + \mathbf{W}_0^{(l)} \mathbf{h}_v^{(l)}\right) \quad (1)$$

The Dependency DAG is discovered among clinical concepts (50 diagnoses, 19 medications, and discretized physiological states) using the PC algorithm (Spirtes et al., 2000) with chi-squared conditional independence testing ($\alpha = 0.01$). Variables are binarized based on clinical presence/absence for diagnoses/medications and clinical thresholds for physiological states (e.g., hypotension defined as MAP<65). The chi-squared test yields 735 directed edges oriented using temporal constraints and Meek rules. For prediction features, we only use edges between pre-onset variables. Edges involving "Sepsis" as a node are discovered separately for offline clinical validation but are *excluded* from both prediction inputs and inference-time explanations to prevent label leakage. For each patient, we compute binary activation vector $\mathbf{a} \in \{0,1\}^{735}$ where edge $k$ is active if both endpoints are in the patient's profile.

### 3.4. Framework Overview

PathwayLLM follows a three-stage architecture (Figure 1). In the first stage, the model encodes each observation window through four specialized encoders capturing temporal physiology, cross-window trajectory, graph-based pa-

tient similarity, and pathway activations. These are fused via cross-modal attention. The second stage integrates encoded representations with a frozen LLM backbone. Four modality-specific soft tokens provide structured context at the input layer, while layer injection reinforces clinical evidence throughout the forward pass. The LLM jointly learns risk prediction and explanation generation. Finally, the third stage aggregates window-level representations into patient-level prediction using Clinical Trajectory LSTM with Deterioration Attention, identifying which windows indicate clinical deterioration.

### 3.5. Stage 1: Multi-Modal Window Encoding

We employ four encoders producing 256-dim representations per window (see Appendix C for parameters).

The Temporal Encoder encodes the 199-dim feature vector through a 3-layer MLP with residual connections:

$$\mathbf{z}_{\text{temp}} = \text{MLP}_3(\text{LayerNorm}(\mathbf{x})) \in \mathbb{R}^{256} \quad (2)$$

The Trajectory Encoder projects diagnosis embeddings to capture patient history context through a 2-layer MLP:

$$\mathbf{z}_{\text{traj}} = \text{MLP}_2(\mathbf{d}) \in \mathbb{R}^{256} \quad (3)$$

where $\mathbf{d} \in \mathbb{R}^{128}$ represents the aggregated representation of the patient's diagnosis history, computed by averaging learned 64-dimensional embeddings of all recorded diagnoses and projecting to 128 dimensions. Cross-window temporal modeling is handled by Stage 3's CT-LSTM, which provides interpretable attention weights identifying critical deterioration windows.

The Graph Encoder leverages a pre-trained 2-layer R-GCN (Schlichtkrull et al., 2018) over the heterogeneous patient-diagnosis-medication graph. To prevent temporal leakage, we use only admission diagnoses and medications administered strictly before the window cutoff time ($t_{end}$). Medications initiated during the prediction horizon ($[t_{end}, t_{end} + 12h]$) are strictly excluded to prevent information leakage from clinician interventions that occur in response to the event we are predicting. This ensures the model relies on the same information state available to a clinician at the moment of prediction. Critically, the R-GCN is pre-trained using only admission diagnoses (recorded at ICU entry, before any potential sepsis onset), so patient node embeddings do not encode post-onset information. Patient similarity edges (via meta-paths) are computed based solely on admission diagnoses, not diagnoses acquired during the ICU stay. For patients present in the training graph, we directly retrieve their learned 64-dimensional embeddings, which are then projected to 256 dimensions via a learned linear layer to match other modalities. For new patients (test set or deployment), we compute inductive embeddings

by aggregating embeddings of their associated diagnosis and medication nodes using the trained R-GCN weights: $\mathbf{z}_{\text{graph}} = \mathbf{W}_{\text{proj}} \cdot \frac{1}{|\mathcal{N}_p^{(k)}|} \sum_{n \in \mathcal{N}_p^{(k)}} \mathbf{h}_n$, where $\mathcal{N}_p^{(k)}$ denotes the set of diagnosis/medication nodes linked to patient $p$ up to window $k$, $\mathbf{h}_n$ are the pre-trained node embeddings, and $\mathbf{W}_{\text{proj}} \in \mathbb{R}^{256 \times 64}$ is the projection matrix.

The Pathway Encoder encodes activated pathways through modulated projection:

$$\mathbf{z}_{\text{path}} = \mathbf{e}_{\text{global}} \odot (1 + \gamma_{\text{path}} \cdot \sigma(\mathbf{W}_{\text{path}} \mathbf{a})) \qquad (4)$$

where $\mathbf{e}_{\text{global}} \in \mathbb{R}^{256}$ is a learnable global pathway prior embedding initialized randomly, $\mathbf{a} \in \{0, 1\}^{735}$ is the binary pathway activation vector, and $\gamma_{\text{path}}$ is a learnable scaling factor. The element-wise modulation allows patient-specific pathway activations to adjust the global prior. The same activated edges used in encoding are also provided to the LLM as natural language pathway descriptions (see Appendix E), ensuring consistency between the pathway signals used for prediction and those presented in generated explanations.

Cross-Modal Fusion integrates the four modalities via multi-head self-attention:

$$\mathbf{z}_{\text{fused}} = \text{CrossModalAttention}(\mathbf{z}_{\text{temp}}, \mathbf{z}_{\text{traj}},$$
$$\mathbf{z}_{\text{graph}}, \mathbf{z}_{\text{path}}) \in \mathbb{R}^{256} \qquad (5)$$

producing attended representations for soft tokens and a fused vector for layer injection.

### 3.6. Stage 2: LLM-Enhanced Prediction

The soft-token and layer-injection pathway is summarized in Figure 3. We evaluate Qwen3-8B (Yang et al., 2025) and LLaMA-3.1-8B (Grattafiori et al., 2024) as backbone LLMs, both with 8 billion parameters.

We implement parameter-efficient fine-tuning using Low-Rank Adaptation (LoRA) (Hu et al., 2022) to adapt the model while preserving pre-trained clinical knowledge. The LLM backbone is kept frozen, and LoRA decomposes weight updates into low-rank matrices to enable efficient fine-tuning with minimal trainable parameters:

$$\mathbf{W}' = \mathbf{W}_0 + \frac{\alpha_{\text{lora}}}{r} \mathbf{B} \mathbf{A} \qquad (6)$$

where $\mathbf{W}_0 \in \mathbb{R}^{d \times d}$ is the frozen pre-trained weight, $\mathbf{A} \in \mathbb{R}^{r \times d}$ and $\mathbf{B} \in \mathbb{R}^{d \times r}$ are trainable low-rank matrices, $r = 16$ is the rank, and $\alpha_{\text{lora}} = 32$ is the scaling factor. We apply LoRA to query (Q) and value (V) projections across all attention layers with dropout 0.1, yielding 8.4M trainable parameters (0.1% of the 8B backbone). This approach allows the model to adapt to sepsis prediction while retaining the LLM's medical reasoning capabilities acquired during pre-training.

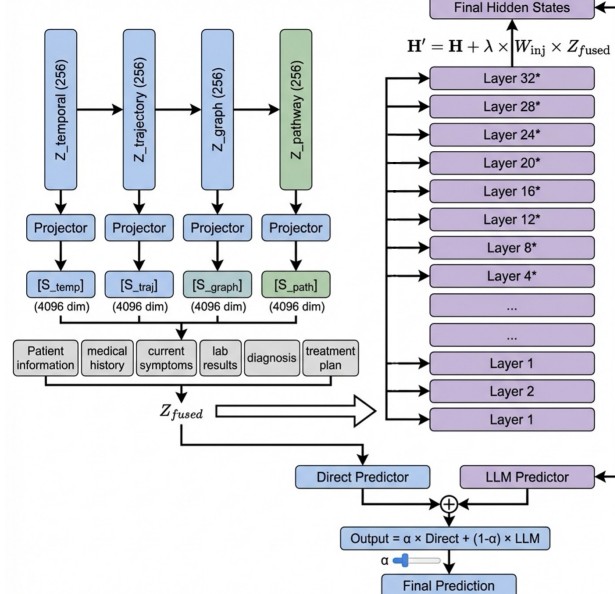

*Figure 3.* Soft Token and Layer Injection. Four modality representations project to 4096-dim soft tokens via 2-layer MLPs, prepended to text. Fused representation injects at layers $\{4, 8, 12, 16, 20, 24, 28, 32\}$ with $\lambda \in [0.2, 0.3]$.

We utilize four modality-specific soft tokens to provide structural context. Each of the four modality representations is independently projected to the LLM hidden dimension via a 2-layer MLP and prepended to the text prompt as learnable soft tokens:

$$\mathbf{s}_m = \text{MLP}_2(\mathbf{z}_m) \in \mathbb{R}^{4096},$$
$$m \in \{\text{temp}, \text{traj}, \text{graph}, \text{path}\} \qquad (7)$$

where each projector uses architecture $256 \rightarrow 512 \rightarrow 4096$ with GELU activation. The complete LLM input sequence is: $[\mathbf{s}_{\text{temp}}; \mathbf{s}_{\text{traj}}; \mathbf{s}_{\text{graph}}; \mathbf{s}_{\text{path}}; \mathbf{e}_1; \ldots; \mathbf{e}_T]$, where $\mathbf{e}_1, \ldots, \mathbf{e}_T$ are the text token embeddings ($T \approx 200\text{–}300$ tokens).

To continuously reinforce clinical evidence throughout the forward pass, we inject the fused representation at 8 intermediate layers selected at stride-4 intervals:

$$\mathbf{H}'^{(l)} = \mathbf{H}^{(l)} + \lambda_l \cdot \text{broadcast}(\mathbf{W}_{\text{inj}}^{(l)} \mathbf{z}_{\text{fused}}) \qquad (8)$$

Injection strength varies by layer depth: $\lambda_l = 0.2$ for shallow layers (4–8), $\lambda_l = 0.25$ for middle layers (12–20), and $\lambda_l = 0.3$ for deep layers (24–32), corresponding to syntactic, semantic, and reasoning processing stages respectively.

We design a structured clinical prompt ($\sim$250 tokens) comprising seven sections: *Patient Demographics* (age, gender, ICU unit type, admission diagnosis); *Hospitalization Progress* (window index, admission time); *Vital Sign Trends* (24h values with directional indicators); *Laboratory Values* (with clinical status annotations); *Current Medications*

(active classes); *Activated Pathways* (natural language descriptions of active edges); and *Task Instruction* (directives for risk assessment and reasoning). See Appendix E for complete prompt template.

We employ two complementary prediction paths that leverage different aspects of the learned representations. The direct predictor operates on fused embeddings: $\hat{y}_{\text{direct}} = \sigma(\mathbf{W}_d \mathbf{z}_{\text{fused}})$. The LLM predictor operates on the final-layer hidden state at the first soft token position: $\hat{y}_{\text{llm}} = \sigma(\mathbf{W}_{\text{llm}} \mathbf{h}_{\text{soft}_1}^{(L)})$. During inference, predictions are fused via a learned combination: $\hat{y} = \alpha_{\text{fuse}} \cdot \hat{y}_{\text{direct}} + (1 - \alpha_{\text{fuse}}) \cdot \hat{y}_{\text{llm}}$, where $\alpha_{\text{fuse}}$ is a learnable scalar parameter that adaptively balances the structured encoding pathway with the LLM's contextual reasoning.

### 3.7. Stage 3: Patient-Level Trajectory Aggregation

After encoding each window through Stages 1–2, we aggregate window-level representations $\{\mathbf{z}_1, \ldots, \mathbf{z}_K\}$ into a patient-level prediction. This stage captures how a patient's condition evolves across their entire ICU stay.

A 2-layer Clinical Trajectory LSTM (CT-LSTM) models temporal evolution across observation windows in a causal manner, ensuring that each window's representation only incorporates information from preceding windows:

$$\mathbf{H} = \text{LSTM}([\mathbf{z}_1, \ldots, \mathbf{z}_K]) \in \mathbb{R}^{K \times 512} \qquad (9)$$

where $\mathbf{h}_k \in \mathbb{R}^{512}$ is the $k$-th row representing the contextualized embedding for window $k$, computed using only windows 1 to $k$. This causal architecture ensures the model can be deployed for real-time prediction without future information leakage. Residual connections preserve original window information: $\mathbf{h}_k' = \text{LayerNorm}(\mathbf{h}_k + \mathbf{W}_{\text{res}} \mathbf{z}_k)$.

We also introduce a Deterioration Attention (DA) mechanism that identifies which windows indicate clinical deterioration. The mechanism combines learned importance scores with temporal position weighting:

$$\alpha_k = \text{softmax}(\mathbf{W}_a^\top \tanh(\mathbf{W}_h \mathbf{h}_k') + \gamma_{\text{time}} \cdot t_k) \qquad (10)$$

where $t_k \in [0, 1]$ is the normalized temporal position (later windows have higher $t_k$) and $\gamma_{\text{time}}$ is a learnable time decay parameter that typically learns positive values, giving higher weight to later windows when deterioration often becomes more apparent. The attention weights $\alpha_k$ provide interpretable insights for clinicians, highlighting which observation windows contributed most to the prediction.

The final patient representation is a weighted sum of window representations, and the prediction is computed via a 2-layer MLP:

$$\mathbf{z}_{\text{patient}} = \sum_{k=1}^{K} \alpha_k \mathbf{h}_k', \quad \hat{y}_{\text{patient}} = \sigma(\text{MLP}(\mathbf{z}_{\text{patient}})) \qquad (11)$$

### 3.8. Synthetic Clinical Reasoning Text Generation

To supervise the LLM branch for explanation generation, we construct synthetic clinical reasoning texts for each observation window using Qwen3-Max (Yang et al., 2025), a state-of-the-art large language model. This approach functions as a knowledge distillation process, transferring the reasoning capabilities of a large-scale model (Qwen3-Max) into our parameter-efficient inference model. We acknowledge that these synthetic reasoning texts represent silver-standard supervision, bounded by the teacher model's accuracy. However, they serve to structure the student model's output format and reasoning process, enforcing consistency with clinical logic (e.g., citing evidence before drawing conclusions) even if they do not introduce novel medical knowledge beyond the teacher's training corpus.

For each observation window, we prompt Qwen3-Max with the patient's clinical features (vital signs, laboratory values, diagnoses, medications) along with contextual information. The prompt instructs the model to generate a concise clinical reasoning explanation that: (1) summarizes the patient's current clinical status; (2) identifies key risk or protective factors observed in the data; (3) describes relevant pathophysiological mechanisms; (4) analyzes activated dependency pathways; and (5) provides a risk assessment.

A multi-stage quality control pipeline ensured consistency and numerical accuracy, with 94% of generated texts passing automated checks (details in Appendix E.1). See Appendix E for the complete generation prompt template.

### 3.9. Training

The training objective combines classification loss $\mathcal{L}_{\text{cls}}$ and language modeling loss $\mathcal{L}_{\text{LM}}$:

$$\mathcal{L} = \mathcal{L}_{\text{cls}} + \lambda_{\text{LM}} \mathcal{L}_{\text{LM}} \qquad (12)$$

The classification loss uses weighted binary cross-entropy to address class imbalance:

$$\mathcal{L}_{\text{cls}} = -\frac{1}{N} \sum_{i=1}^{N} [w_+ y_i \log(\hat{y}_i) + (1 - y_i) \log(1 - \hat{y}_i)] \qquad (13)$$

where $w_+ = 1/p_+ \approx 11.8$ is the inverse frequency weight for the positive class (sepsis prevalence $p_+ = 8.45\%$).

The language modeling loss ensures the LLM branch generates evidence-conditioned explanations supervised by Qwen3-Max-generated clinical reasoning texts:

$$\mathcal{L}_{\text{LM}} = -\frac{1}{T} \sum_{t=1}^{T} \log P(r_t \mid r_{<t}, \mathbf{s}_{1:4}, \text{prompt}) \qquad (14)$$

where $r_t$ are reasoning text tokens and $\mathbf{s}_{1:4}$ are the four soft tokens. During inference, the model generates explanations

conditioned only on encoded features without access to ground truth labels.

We employ a two-phase training strategy. Phase 1 trains Stages 1–2 (encoders, LLM adapters) for 15 epochs using window-level labels $y_k^{(w)}$ for classification and reasoning text supervision, with encoder learning rate $10^{-4}$ and LoRA learning rate $10^{-5}$. Phase 2 freezes Stages 1–2 and trains Stage 3 (CT-LSTM and DA) for 5 epochs using patient-level labels $y^{(p)}$, enabling patient-level aggregation to benefit from stable window representations. We use the AdamW optimizer with cosine annealing ($\lambda_{\text{LM}} = 0.5$, gradient clipping 1.0, bfloat16 precision). The model has a total of 29.9M trainable parameters. See Appendix D for complete hyperparameters.

## 4. Results

### 4.1. Main Performance Comparison

Table 1 presents comprehensive comparison on the MIMIC-IV test set (2,312 patients, 196 with sepsis). PathwayLLM with Qwen3-8B achieves patient-level AUROC 0.891 (95% CI: 0.872–0.910; bootstrap, n=1000) and AUPRC 0.724 (95% CI: 0.687–0.761), representing absolute improvements of 0.070 and 0.196 over the best baseline (COMPOSER).

Baselines span traditional ML models, neural time-series models (TCN (Bai et al., 2018), Transformer (Vaswani et al., 2017)), GAT (Veličković et al., 2018) applied to the EHR graph, structured EHR representation models, and clinical sepsis systems. These methods achieve AUROC in the 0.68–0.82 range, with COMPOSER being the strongest baseline. However, all baselines show substantially lower AUPRC (0.20–0.53), indicating difficulty in handling the class-imbalanced nature of sepsis prediction. Our model's AUPRC of 0.724 represents a 37% relative improvement over COMPOSER, demonstrating superior performance at detecting positive cases. Additional irregular time-series baselines are reported in Appendix I.

### 4.2. External Validation

We evaluated cross-institutional generalization on eICU v2.0 (Pollard et al., 2018), using equivalent inclusion criteria that yielded 12,847 patients from 143 hospitals with 9.2% sepsis prevalence. Because eICU uses different coding systems, APACHE diagnoses and medication names were mapped to ICD-10 and ATC categories; unmapped dependency edges were treated as inactive. As shown in Table 2, zero-shot transfer from MIMIC-IV to eICU achieved AUROC 0.842 and AUPRC 0.651, while fine-tuning only Stage 3 on 20% of eICU patients improved performance to 0.867/0.689. See Appendix H for mapping details and full baseline results.

*Table 1.* Performance comparison on MIMIC-IV test set (2,312 patients). All methods were trained and evaluated on the identical suspected-infection cohort. Results reported as mean $\pm$ std over 5 random seeds. All methods use identical training setup including weighted cross-entropy loss ($w_+ = 11.8$) and patient-level evaluation; baselines use max-pooling for aggregation.

| METHOD | AUROC | AUPRC | SE (SP=.9) | F1 |
|---|---|---|---|---|
| LOGISTIC REG. | $.682_{\pm.021}$ | $.198_{\pm.018}$ | $.321_{\pm.024}$ | $.243_{\pm.019}$ |
| RANDOM FOREST | $.721_{\pm.019}$ | $.267_{\pm.021}$ | $.384_{\pm.026}$ | $.302_{\pm.022}$ |
| XGBOOST | $.742_{\pm.018}$ | $.312_{\pm.023}$ | $.417_{\pm.025}$ | $.341_{\pm.021}$ |
| LSTM | $.758_{\pm.020}$ | $.368_{\pm.028}$ | $.452_{\pm.031}$ | $.378_{\pm.025}$ |
| GRU | $.763_{\pm.019}$ | $.381_{\pm.027}$ | $.467_{\pm.030}$ | $.389_{\pm.024}$ |
| TCN | $.781_{\pm.017}$ | $.423_{\pm.025}$ | $.513_{\pm.028}$ | $.421_{\pm.023}$ |
| TRANSFORMER | $.794_{\pm.016}$ | $.458_{\pm.024}$ | $.536_{\pm.027}$ | $.447_{\pm.022}$ |
| GAT | $.786_{\pm.018}$ | $.441_{\pm.026}$ | $.503_{\pm.029}$ | $.423_{\pm.024}$ |
| GRASP | $.793_{\pm.017}$ | $.462_{\pm.024}$ | $.524_{\pm.027}$ | $.448_{\pm.022}$ |
| MED-BERT | $.802_{\pm.016}$ | $.473_{\pm.023}$ | $.532_{\pm.026}$ | $.458_{\pm.021}$ |
| CEHR-BERT | $.805_{\pm.015}$ | $.481_{\pm.022}$ | $.548_{\pm.025}$ | $.467_{\pm.020}$ |
| NYUTRON | $.814_{\pm.014}$ | $.506_{\pm.021}$ | $.576_{\pm.024}$ | $.498_{\pm.019}$ |
| COMPOSER | $.821_{\pm.014}$ | $.528_{\pm.020}$ | $.602_{\pm.023}$ | $.512_{\pm.018}$ |
| PATHWAYLLM (LLAMA) | $.876_{\pm.012}$ | $.698_{\pm.016}$ | $.723_{\pm.019}$ | $.671_{\pm.015}$ |
| **PATHWAYLLM (QWEN)** | $\mathbf{.891}_{\pm.011}$ | $\mathbf{.724}_{\pm.015}$ | $\mathbf{.751}_{\pm.018}$ | $\mathbf{.698}_{\pm.014}$ |

*Table 2.* External validation on eICU. Zero-shot applies the MIMIC-IV-trained model directly; fine-tuned adapts Stage 3 using 20% eICU data.

| SETTING | AUROC | AUPRC |
|---|---|---|
| MIMIC-IV TEST | .891 | .724 |
| EICU ZERO-SHOT | .842 | .651 |
| EICU FINE-TUNED (20%) | .867 | .689 |

### 4.3. Ablation Studies

Table 3 shows systematic ablation results revealing the contribution of each component. The multi-modal encoding strategy proves essential: R-GCN contributes significantly ($\Delta$AUROC = -3.7%), confirming that population-level patient similarity patterns provide essential signals unavailable from individual trajectories alone. The Temporal Encoder (-2.2%) and Trajectory Module (-1.8%) demonstrate the importance of both within-window feature extraction and cross-window temporal modeling. The Pathway Encoder contributes to both prediction (0.9%) and interpretability, with its primary value in providing mechanistic explanations that clinicians find intuitive.

The LLM integration experiments reveal that auxiliary embeddings (-3.3%) contribute more than layer injection (-0.7%), suggesting that providing structured context at the input level is more effective than intermediate feature injection. The comparison between four modality-specific soft tokens and single-token fusion (+1.2%) validates our design choice of preserving modality-specific information rather than collapsing all signals into a single representation.

Stage 3 components (CT-LSTM and DA) collectively con-

*Table 3.* Ablation results on MIMIC-IV test set. AUC=AUROC, PRC=AUPRC. "w/o LLM" removes the entire Stage 2 (soft token projectors, layer injection, LoRA adapters), using only the direct predictor on fused Stage 1 embeddings. "w/o Aux. Embed." removes soft tokens while keeping layer injection.

| Variant | AUC | PRC | Δ |
|---|---|---|---|
| Full (Qwen3-8B) | .891 | .724 | — |
| *Encoders* | | | |
| w/o R-GCN | .854 | .642 | -3.7% |
| w/o Temporal | .869 | .682 | -2.2% |
| w/o Trajectory | .873 | .695 | -1.8% |
| w/o Pathway | .882 | .712 | -0.9% |
| *LLM Integration* | | | |
| w/o LLM | .867 | .679 | -2.4% |
| w/o Layer Inj. | .884 | .712 | -0.7% |
| w/o Aux. Embed. | .858 | .652 | -3.3% |
| Single Token | .879 | .698 | -1.2% |
| *Training* | | | |
| w/o Synth. Sup. | .858 | .584 | -3.3% |
| w/o Class Wt. | .812 | .421 | -7.9% |
| *Alt. Backbone* | | | |
| LLaMA-3.1-8B | .884 | .698 | -0.7% |
| *Stage 3* | | | |
| w/o CT-LSTM | .849 | .631 | -4.2% |
| w/o DA | .874 | .691 | -1.7% |

*Table 4.* Window-level and patient-level performance on the MIMIC-IV test set.

| Evaluation | AUROC | AUPRC |
|---|---|---|
| Window, Stages 1–2 | .843 | .406 |
| Patient, max pooling | .849 | .631 |
| Patient, full model | .891 | .724 |

patients had fewer windows than negatives due to onset censoring ($3.2\pm1.4$ vs. $4.8\pm2.6$), but a logistic regression using only window count achieved AUROC 0.612. Removing the explicit temporal-position input from DA retained AUROC 0.884, and window-count-stratified evaluation remained stable across strata (AUROC 0.864–0.896; Appendix L). Longer prediction horizons and graph-construction sensitivity analyses are reported in Appendices J and K. These analyses suggest that the model is not primarily exploiting length or time-position artifacts.

### 4.5. Case Studies and Clinical Evaluation

See Appendix F for detailed representative case studies illustrating model behavior, including true positive deterioration trajectories and false positive examples.

Two ICU physicians independently evaluated 50 randomly selected test cases (25 TP, 25 TN) in a blinded manner, without knowledge of ground truth labels or model predictions during the assessment. Evaluation criteria included: reasoning coherence (1–5 scale), risk factor accuracy, numerical accuracy, pathway validity, and clinical actionability. The evaluation yielded a mean reasoning coherence of 4.2/5.0, risk factor accuracy of 89%, numerical accuracy of 94%, pathway validity of 82%, and clinical actionability of 3.9/5.0, with an inter-rater agreement of $\kappa$=0.73. Physicians noted explanations were most valuable for identifying non-obvious risk factors. Representative dependency pathways are listed in Appendix G. Beyond discrimination, the model demonstrates good probability calibration (Brier score 0.048, ECE 0.019; see Appendix M), supporting clinical deployment where reliable probability estimates are essential.

While overall coherence was high, our clinician review identified two primary failure patterns which future work must address: (1) Hallucinated Specificity: In 6% of cases, the model generated specific values (e.g., "Lactate 4.5") when the input feature was present but the exact value was missing or different. This suggests the language model occasionally prioritizes plausible-sounding narrative over faithful data copying. (2) Over-Attribution: In 11% of false positives, the model correctly identified abnormal vitals (e.g., hypotension) but incorrectly attributed them to sepsis rather than alternative causes like heart failure, reflecting the model's single-task training bias.

tribute 5.9% improvement, with CT-LSTM alone accounting for 4.2%. This demonstrates that patient-level trajectory aggregation is essential for capturing clinical deterioration patterns. Notably, even without Stage 3 (w/o CT-LSTM), the model achieves 0.849 AUROC, still significantly outperforming the best baseline (COMPOSER: 0.821), confirming that our multi-modal encoding and LLM enhancement provide substantial gains independent of the aggregation strategy. Training strategy ablations confirm that class weighting and synthetic reasoning supervision both contribute to final performance.

### 4.4. Robustness Analyses

Beyond component ablations, we conducted additional analyses to verify that the observed gains do not arise from aggregation artifacts or temporal shortcuts.

Table 4 further separates window-level prediction from patient-level aggregation. On 10,751 test windows (255 positives; 2.37% prevalence), Stages 1–2 achieve AUROC 0.843 and AUPRC 0.406. Max-pooling these window predictions yields patient-level AUROC/AUPRC 0.849/0.631, while CT-LSTM aggregation improves to 0.891/0.724, indicating that gains arise from both strong window representations and trajectory modeling.

We also tested whether performance could be explained by stay-length or temporal-position shortcuts. Sepsis-positive

## 5. Discussion

The R-GCN graph encoder emerged as the most influential component (3.7% AUROC improvement when ablated), validating that patients with similar diagnosis-medication profiles share sepsis risk patterns that purely sequential models cannot capture. Pathway signals from dependency discovery contribute modestly to prediction (0.9%) but provide mechanistic interpretability that physicians find more intuitive than correlation-based attribution methods. The parameter-efficient strategy (29.9M trainable parameters, 0.4% of frozen 8B LLM) enables deployment on standard GPU infrastructure. Inference required approximately 0.3s per window and 1.5s per patient on a single A800 GPU, compatible with the 12-hour clinical update cycle.

Several limitations warrant discussion. Our model targets risk stratification within suspected-infection populations rather than general ICU screening, requiring prior clinical suspicion of infection before application. The current framework uses only structured EHR features, excluding clinical notes, imaging, and other unstructured data sources that may contain additional predictive information. While we provide retrospective evaluation on MIMIC-IV with external validation on eICU, prospective validation in real clinical workflows, including alert integration, clinician feedback loops, and assessment of downstream patient outcomes, is needed before deployment. Finally, we did not perform fairness analysis across demographic subgroups such as age, sex, and race. Because clinical models often exhibit performance disparities across demographic groups and sepsis diagnosis may reflect systemic biases in care delivery, rigorous fairness auditing is needed before clinical deployment.

## 6. Conclusion

We introduced PathwayLLM, a trajectory-based framework that grounds patient-level sepsis prediction on structured clinical evidence. The framework integrates multi-view encoding (temporal, graph, and dependency-pathway signals) with Clinical Trajectory LSTM and Deterioration Attention for patient-level aggregation, while jointly generating evidence-conditioned text explanations. On MIMIC-IV, PathwayLLM achieves AUROC 0.891 and AUPRC 0.724 with clinician-validated explanation quality, and eICU validation supports cross-institutional transfer. Our results establish foundations for trustworthy clinical AI systems combining predictive accuracy with interpretable, evidence-grounded outputs.

## Acknowledgements

This work was supported by the National Natural Science Foundation of China (Grant No. 62372391), Fujian Provincial Major Science and Technology Project (Grant No. 2022YZ040011), and National Key Research and Development Program of China (Grant Nos. 2024YFF1206204, 2023YFF1205602).

## Impact Statement

This paper presents work advancing Machine Learning for healthcare. We develop a sepsis prediction system that could improve early detection of this life-threatening condition. We emphasize interpretability as particularly important in clinical AI, where clinicians must understand the reasoning behind predictions to verify plausibility and take appropriate action. Our framework addresses this by generating evidence-conditioned explanations alongside risk scores. Although the model was trained primarily on MIMIC-IV and externally validated on eICU, prospective validation across diverse clinical workflows remains necessary before deployment. The system is designed to assist, not replace, clinical decision-making.

## Ethics and Data Use Statement

This study used de-identified public critical care databases, MIMIC-IV v3.1 and eICU v2.0, under the required PhysioNet credentialing, training, and data use agreements. Patient consent for these databases was handled by the original data stewards and waived for secondary use of de-identified data. The study protocol was approved by the Ethics Review Committee of The Second Affiliated Hospital of Zhejiang Chinese Medical University (Approval No. 2026-087-01). The clinician evaluation involved review of de-identified case summaries and model outputs only, with no patient contact, intervention, or access to identifiable patient information.

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

## A. Cohort Selection Details

The cohort selection process is detailed in Figure 4. We applied sequential exclusion criteria to the initial 94,458 ICU stays in MIMIC-IV to identify the final study population.

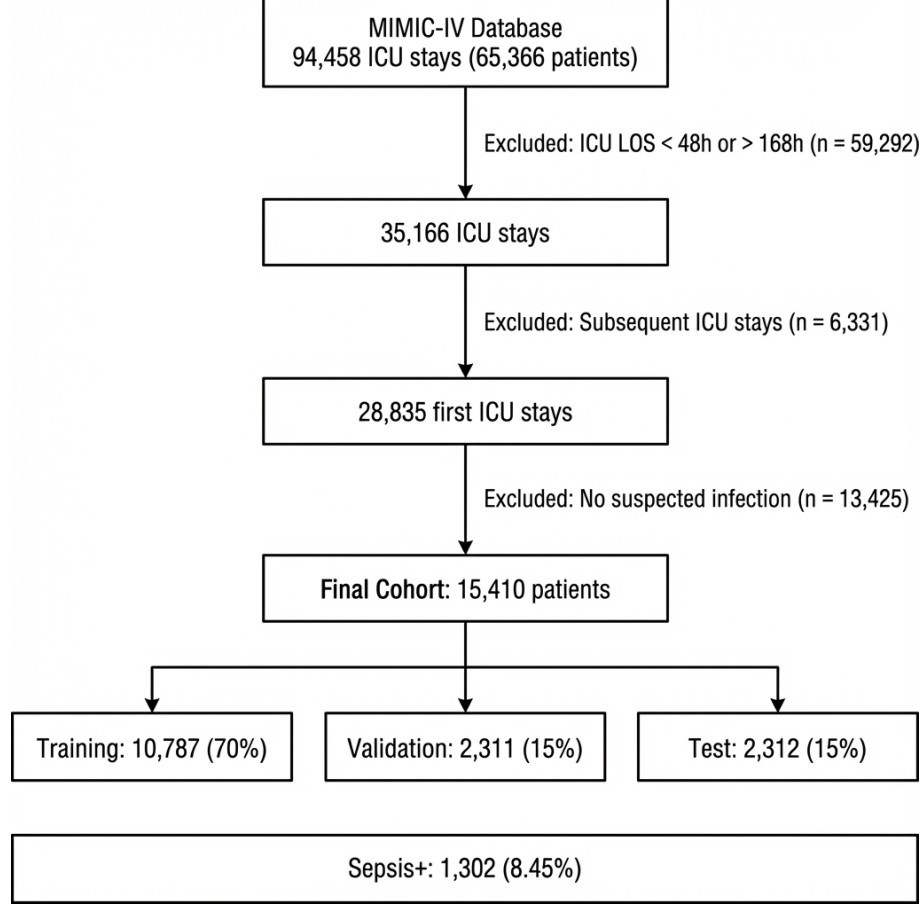

*Figure 4.* Patient Selection Flowchart. Starting from 94,458 ICU stays in MIMIC-IV, we applied sequential exclusion criteria. The final cohort of 15,410 unique adult patients generated 71,724 observation windows.

Sepsis was defined according to Sepsis-3 criteria. *Suspected infection time* was defined as the earlier of antibiotic administration or blood culture sampling, where culture occurs within −24 to +72 hours of antibiotic. *Sepsis onset time* was defined as the first time point when SOFA increase $\geq 2$ within $\pm 48$h of suspected infection.

**Preprocessing and Feature Engineering.** We employed a two-stage imputation approach designed to preserve temporal dynamics while handling sparse laboratory measurements (missing rates: heart rate 2.1%, BP 3.4%, SpO$_2$ 3.1%, temperature 8.7%, GCS 12.4%, WBC 18.3%, lactate 42.8%, blood gas 45.3%). In the first stage (forward-fill), missing values within each patient stay were addressed using last observation carried forward (LOCF) with a maximum gap of 4 hours for vital signs. In the second stage, remaining missing values were imputed using clinical defaults or population medians: SpO$_2$ with 98%, temperature with 37.0°C, GCS with 15, lactate with 1.5 mmol/L, and others with training set medians/normal values. Physiologically implausible values were identified and corrected using clinical range constraints: heart rates outside 20–300 bpm, blood pressures outside 30–300 mmHg (systolic) or 10–200 mmHg (diastolic), temperatures outside 25–45°C, and SpO$_2 > 100\%$ were flagged or capped. Extreme laboratory outliers beyond 3 interquartile ranges were winsorized to the 1st/99th percentiles. Following imputation and outlier handling, all continuous features were standardized to zero mean and unit variance using training set statistics. Finally, for each 24-hour observation window, measurements were aligned to the window start time, with statistics (mean, std, min, max, last) computed from all measurements within the window.

## B. Complete Feature List

*Vital Signs:* Heart rate, SBP, DBP, MAP, respiratory rate, SpO$_2$, temperature, GCS; each includes 5 statistics (mean, std, min, max, last).

*Laboratory Values:* CBC (WBC, hemoglobin, platelets), metabolic panel (glucose, BUN, creatinine, sodium, potassium, chloride, bicarbonate, anion gap), blood gas (pH, PaO$_2$, PaCO$_2$, lactate), coagulation (INR); each includes 5 statistics.

*Diagnosis Categories:* ICD-10-CM categories mapped from first 3 characters, covering infectious diseases, cardiovascular (8 subcategories), respiratory (6 subcategories), and others.

*Medication Categories:* ATC Level 3 groups including antibiotics, vasopressors, sedatives, opioids, anticoagulants, insulin, corticosteroids, diuretics.

## C. Model Architecture Details

The trainable parameter distribution across the three stages of PathwayLLM is summarized as follows (Total Trainable: 29.9M; Frozen LLM Backbone: 8B). The majority of trainable parameters (85%) are allocated to the LLM adaptation layers in Stage 2: Soft Token Projectors (8.9M), Layer Injection (8.4M), and LoRA Adapters (8.4M). In contrast, the core structural encoders in Stage 1 remain lightweight: Temporal Encoder (117K), Trajectory Encoder (66K), Graph Encoder (820K, R-GCN), Pathway Encoder (189K), and Cross-Modal Fusion (918K). Stage 3 components include the CT-LSTM (1.6M), Deterioration Attention (395K), and Patient-Level Classifier (66K).

## D. Training Hyperparameters

Complete training hyperparameters are provided in Table 5. We used a cosine annealing schedule and gradient accumulation to stabilize training of the multi-modal components.

*Table 5.* Training hyperparameters.

| Hyperparameter | Value |
|---|---|
| Epochs | 20 |
| Batch size | 4 (gradient accumulation: 8) |
| Effective batch size | 32 |
| Encoder learning rate | 1e-4 |
| LoRA learning rate | 1e-5 |
| Optimizer | AdamW ($\beta_1$=0.9, $\beta_2$=0.999) |
| Scheduler | Cosine annealing with 500 warmup steps |
| Gradient clipping | 1.0 |
| Precision | bfloat16 |
| Classification weight ($w_+$) | $\approx$11.8 (= 1/0.0845) |
| LM loss weight ($\lambda_{LM}$) | 0.5 |
| LoRA rank ($r$) | 16 |
| LoRA scaling ($\alpha_{\text{lora}}$) | 32 |
| LoRA target modules | q_proj, v_proj* |
| Random seeds | 42, 123, 456, 789, 2024 |

*Following Hu et al. (2022), we apply LoRA to query and value projections.

*Computational Resources:* 2$\times$NVIDIA A800 80GB GPUs, AMD EPYC 7763 CPU (64 cores), 512GB RAM.

*Computational cost.* Training Stages 1–2 required approximately 18 hours on 2$\times$A800 80GB GPUs, while Stage 3 required approximately 2 hours. At inference time, processing required approximately 0.3s per observation window and 1.5s per patient on a single A800 GPU. The model has 29.9M trainable parameters, including 8.4M LoRA parameters.

# E. Prompt Template Design

The prompt template provides structured clinical context to the LLM beyond the auxiliary contextual embeddings. The template follows a consistent format across all patient predictions, comprising seven major sections that progressively present information from general context to specific clinical details.

The system section establishes the LLM's role as a clinical decision support system specializing in sepsis risk assessment, instructing it to analyze patient data and provide structured clinical reasoning. The patient information section presents demographic details including age, gender, unit type, primary admission diagnosis, and current ICU day.

The vital signs section reports the most recent 24-hour window of physiological measurements. For each vital sign (heart rate, blood pressure, mean arterial pressure, respiratory rate, temperature, oxygen saturation, and Glasgow Coma Scale), the prompt includes the mean value, observed range (minimum to maximum), and directional trend indicator using arrows: $\uparrow$ for rising values, $\downarrow$ for declining values, and $\rightarrow$ for stable measurements.

The laboratory values section presents the most recent available results for key sepsis-relevant measurements: white blood cell count, hemoglobin, platelet count, lactate, creatinine, and bilirubin. Each value is accompanied by a clinical status annotation categorizing it as normal, low, high, or critical based on standard reference ranges.

The medications and diagnoses sections enumerate the patient's active medication classes and documented diagnoses, providing clinical context for the pathway interpretation. The active pathways section describes which discovered dependency edges are currently active, expressed in natural language (for example, "Pneumonia leading to systemic inflammatory response"). Critically, edges involving "Sepsis" as a node (e.g., Sepsis→Vasopressors) are excluded from both prediction input and inference-time explanations to prevent outcome-related information leakage.

The instruction section directs the LLM to assess window-level 12-hour sepsis onset risk, which will be aggregated with other windows via Clinical Trajectory LSTM to form patient-level predictions. The response is structured around five components: identification of risk factors suggesting infection or organ dysfunction, active dependency mechanisms and pathophysiological pathways, temporal trend analysis of how the condition is evolving, a numerical probability estimate with supporting reasoning, and suggested further assessments or monitoring considerations for clinical team discussion (non-prescriptive, assistive only).

### E.1. Quality Control Pipeline for Synthetic Reasoning

We implement a multi-stage quality control pipeline to ensure reasoning text quality. First, we enforce directional consistency: positive samples must emphasize risk factors (e.g., elevated lactate, hypotension), while negative samples must identify protective factors, rejecting texts with contradictory conclusions. Second, we verify numerical accuracy by ensuring clinical values referenced in the text match actual patient data within strict tolerances ($\pm 5$ for vital signs, $\pm 0.5$ for laboratory values). Third, length validation ensures reasoning texts contain 100–200 words. Rejected samples are regenerated up to 3 times, and 5% of generated texts are randomly sampled for expert clinician review to ensure clinical plausibility.

# F. Additional Case Studies

*Case 1 (True Positive):* A 72-year-old male with diabetes and pneumonia showed progressive deterioration: risk escalated from 0.42 to 0.78 as MAP declined (95→68 mmHg) and lactate increased (1.4→2.1 mmol/L). The DA mechanism appropriately weighted later windows, and sepsis onset occurred within 4 hours of the final prediction. *Case 2 (True Negative):* A 54-year-old female with UTI showed risk decrease (0.48→0.23) as inflammatory markers improved under antibiotics. *Case 3 (False Positive):* A 68-year-old male with acute pancreatitis presented with tachycardia (HR 105), fever (38.1°C), and WBC 16.2 K/$\mu$L. The model predicted 0.72 risk, identifying the "Pancreatitis → SIRS" pathway as a major contributor. The patient did not develop sepsis, as the SIRS was from sterile pancreatitis. This case illustrates that the model cannot distinguish infectious from non-infectious SIRS.

*Case 4 (False Negative):* A 61-year-old immunocompromised female (renal transplant) had a subtle presentation: temperature 37.6°C, HR 78, WBC 4.2 K/$\mu$L (leukopenic). The model assigned 0.31 patient-level risk. The patient developed CMV reactivation with sepsis at 18 hours after the last observation window. This case highlights that blunted inflammatory response in immunocompromised patients leads to atypical presentations that current features cannot capture.

## G. Discovered Dependency Pathways

The following edges represent structural dependencies discovered via the PC algorithm from training data. Table 6 lists the most frequent edges in sepsis-positive patients, while Table 7 provides additional discovered edges. For prediction, only edges involving pre-onset variables (diagnoses, medications administered before the prediction time) are used as input features. Edges involving "Sepsis" or post-onset treatments (e.g., vasopressors initiated after sepsis onset) are included in this table for interpretability and clinical validation purposes, but are *not* used as input features for prediction windows.

*Table 6.* Top 10 dependency edges by frequency in sepsis-positive patients (Retrospective validation only; excluded from model input/explanation).

| Edge | Freq. | Clinical Interpretation |
|---|---|---|
| Bacteremia → Sepsis | 89.4% | Disease progression |
| Sepsis → Vasopressors | 71.3% | Hemodynamic support |
| Resp. Failure → Mech. Vent. | 62.1% | Respiratory support |
| Infection → Fever | 45.8% | Inflammatory response |
| Diabetes → Infection | 35.2% | Risk factor |
| Pneumonia → Antibiotics | 34.2% | Standard treatment |
| AKI → Sepsis | 31.4% | Organ dysfunction |
| Heart Failure → Diuretics | 29.8% | Standard treatment |
| UTI → Antibiotics | 28.7% | Standard treatment |
| COPD → Pneumonia | 22.8% | Comorbidity link |

*Table 7.* Additional discovered dependency edges.

| Edge | Freq. | Clinical Interpretation |
|---|---|---|
| Neoplasm → Sepsis | 15.6% | Immunosuppression risk |
| Liver Disease → Coagulopathy | 18.2% | Synthesis dysfunction |
| CKD → Hyperkalemia | 22.4% | Renal clearance |
| Hypotension → Lactate | 55.3% | Hypoperfusion |
| Tachycardia → Beta-Blocker | 41.2% | Rate control |
| Fever → Antipyretics | 38.9% | Symptomatic relief |

## H. External Validation on eICU

To evaluate generalizability beyond the primary MIMIC-IV cohort, we conducted external validation on the eICU Collaborative Research Database v2.0 (Pollard et al., 2018), a multi-center critical care database containing records from 208 hospitals across the United States.

We validated on the eICU Collaborative Research Database, applying equivalent inclusion criteria (adults, ICU 48–168h, first admission, suspected infection) which yielded 12,847 patients from 143 hospitals with 9.2% sepsis prevalence. As eICU uses different coding systems, we mapped APACHE diagnoses to ICD-10 and drug names to ATC Level 3 categories using semantic matching. Concepts not in the MIMIC-IV vocabulary were mapped to the closest equivalents for the HIN encoder, while dependency edges involving unmapped concepts were treated as inactive. We evaluated two transfer scenarios: zero-shot transfer (applying the MIMIC-IV-trained model directly) and fine-tuned transfer (adapting Stage 3 aggregation layers on 20% of eICU data).

Table 8 presents the transfer learning results. In the zero-shot setting, PathwayLLM maintains robust performance (AUROC 0.842), significantly outperforming baselines and COMPOSER (0.756). The 4.9% performance gap compared to MIMIC-IV is attributable to coding differences (proprietary vs. ICD-10), sparser vital sign documentation in eICU, and population heterogeneity from community hospitals. Fine-tuning Stage 3 recovers 2.5% AUROC (to 0.867), suggesting that while temporal aggregation patterns differ across institutions, the fundamental multi-modal representations transfer well. These results support the model's potential for multi-site deployment, exceeding traditional scoring systems (e.g., qSOFA) even without adaptation.

*Table 8.* External validation performance on eICU dataset. Zero-shot = MIMIC-IV model applied directly; Fine-tuned = adapted with 20% eICU data; Trained = trained from scratch on eICU.

| Method | AUROC | AUPRC | Se (Sp=.9) | Sp (Se=.9) |
|---|---|---|---|---|
| *Zero-shot Transfer (MIMIC-IV to eICU)* | | | | |
| XGBoost | 0.698 | 0.267 | 0.352 | 0.423 |
| LSTM | 0.712 | 0.298 | 0.378 | 0.456 |
| Transformer | 0.728 | 0.324 | 0.401 | 0.478 |
| COMPOSER | 0.756 | 0.412 | 0.468 | 0.521 |
| PathwayLLM (LLaMA) | 0.823 | 0.618 | 0.492 | 0.587 |
| PathwayLLM (Qwen) | 0.842 | 0.651 | 0.518 | 0.612 |
| *Fine-tuned Transfer (20% eICU data)* | | | | |
| PathwayLLM (Qwen) | 0.867 | 0.689 | 0.574 | 0.658 |
| *Trained on eICU (100% eICU data)* | | | | |
| XGBoost | 0.734 | 0.305 | 0.408 | 0.487 |
| LSTM | 0.793 | 0.542 | 0.421 | 0.534 |
| Transformer | 0.811 | 0.578 | 0.456 | 0.562 |
| COMPOSER | 0.824 | 0.596 | 0.523 | 0.598 |

## I. Additional Irregular Time-Series Baselines

We further evaluated recent irregular time-series methods that operate directly on raw timestamped ICU observations, including mTAND (Shukla & Marlin, 2021), Raindrop (Zhang et al., 2022), and DuETT (Labach et al., 2023). Table 9 shows that these baselines perform comparably to the strongest methods in Table 1 but remain below PathwayLLM.

*Table 9.* Additional irregular time-series baselines on the MIMIC-IV test set.

| Method | AUROC | AUPRC | F1 |
|---|---|---|---|
| mTAND | 0.801±0.017 | 0.478±0.024 | 0.452±0.022 |
| Raindrop | 0.812±0.016 | 0.502±0.023 | 0.481±0.021 |
| DuETT | 0.818±0.015 | 0.519±0.022 | 0.496±0.020 |

## J. Prediction Horizon Analysis

We extended the prediction horizon from 12h to 24h and 48h by redefining window-level labels. Table 10 shows that performance degrades gracefully at longer horizons, as expected for earlier prediction based on subtler physiological signals.

*Table 10.* Performance across prediction horizons on the MIMIC-IV test set.

| Horizon | AUROC | AUPRC | Se (Sp=.9) |
|---|---|---|---|
| 12h | 0.891 | 0.724 | 0.751 |
| 24h | 0.861 | 0.672 | 0.698 |
| 48h | 0.823 | 0.598 | 0.634 |

## K. Dependency Graph Robustness

We evaluated sensitivity to dependency graph construction by varying the PC algorithm's significance threshold. Table 11 shows stable performance across graph densities. As an algorithmic check, replacing PC with Greedy Equivalence Search (GES) yielded AUROC 0.888 and AUPRC 0.719.

*Table 11.* PC threshold sensitivity on the MIMIC-IV test set.

| PC threshold $\alpha$ | Edges | AUROC | AUPRC |
|---|---|---|---|
| 0.001 | 523 | 0.889 | 0.721 |
| 0.01 | 735 | 0.891 | 0.724 |
| 0.05 | 1,283 | 0.887 | 0.718 |
| 0.10 | 1,891 | 0.883 | 0.711 |

## L. Robustness Analyses

Table 12 reports window-count-stratified evaluation. Within each stratum, window count cannot discriminate sepsis-positive from sepsis-negative patients, yet AUROC remains stable from 0.864 to 0.896.

*Table 12.* Window-count-stratified performance on the MIMIC-IV test set.

| Windows | Sepsis+ | Sepsis- | AUROC |
|---|---|---|---|
| 2–3 | 82 | 524 | 0.864 |
| 4–5 | 68 | 712 | 0.878 |
| 6–7 | 31 | 487 | 0.889 |
| 8+ | 15 | 393 | 0.896 |

## M. Calibration Analysis

Beyond discrimination (AUROC/AUPRC), clinical deployment requires well-calibrated probability estimates. PathwayLLM with Qwen3-8B achieves Brier score 0.048 (vs. COMPOSER 0.055), Expected Calibration Error (ECE) 0.019, and Maximum Calibration Error (MCE) 0.043, indicating well-calibrated predictions (Table 13). The LLaMA variant shows slightly higher but still competitive calibration (Brier 0.051, ECE 0.022). The improvement in calibration is consistent with the model's multi-view evidence integration, which provides more grounded probability estimates.

*Table 13.* Calibration metrics on MIMIC-IV test set. Lower Brier score and ECE indicate better calibration.

| Method | Brier Score | ECE | MCE |
|---|---|---|---|
| Logistic Regression | 0.078 | 0.042 | 0.089 |
| Random Forest | 0.074 | 0.039 | 0.085 |
| XGBoost | 0.071 | 0.038 | 0.082 |
| LSTM | 0.065 | 0.035 | 0.076 |
| GRU | 0.064 | 0.034 | 0.074 |
| TCN | 0.062 | 0.032 | 0.071 |
| Transformer | 0.061 | 0.031 | 0.068 |
| GAT | 0.063 | 0.033 | 0.072 |
| GRASP | 0.060 | 0.030 | 0.067 |
| Med-BERT | 0.059 | 0.029 | 0.065 |
| CEHR-BERT | 0.058 | 0.028 | 0.063 |
| NYUTron | 0.056 | 0.027 | 0.059 |
| COMPOSER | 0.055 | 0.026 | 0.057 |
| PathwayLLM (LLaMA) | 0.051 | 0.022 | 0.047 |
| PathwayLLM (Qwen) | **0.048** | **0.019** | **0.043** |

We further assessed calibration by binning predicted probabilities into 10 equal-width bins and computing observed sepsis rates within each bin. The calibration curve shows good agreement between predicted and observed risks, with slight underestimation in the highest risk decile (predicted 0.8–1.0: observed 0.76). This conservative behavior may be clinically

appropriate as it reduces alert fatigue while maintaining high sensitivity. Regarding clinical decision thresholds, at the default threshold of 0.5, the model achieves PPV 0.421 and NPV 0.978. For high-sensitivity deployment (threshold 0.3), sensitivity increases to 0.89 with PPV 0.287, corresponding to approximately 3.5 alerts per true positive. This performance is comparable to existing sepsis alert systems in clinical practice.

