# OpenReview forum: "PathwayLLM: Explainable Clinical Trajectory Modeling with Structured Pathways for Sepsis Prediction"
_ICML.cc/2026/Conference — ICML 2026 regular_

### Official Review · Reviewer_F1Jk · 2026-03-10

**Soundness:** 2
**Presentation:** 3
**Significance:** 3
**Originality:** 3
**Overall Recommendation:** 4
**Confidence:** 4

**Summary:**

This paper proposes PathwayLLM for patient-level sepsis prediction from ICU EHR trajectories. Each 24 hour observation window is encoded from four multi-views and then integrated into an LLM through soft tokens and layer injection, and finally aggregated across windows with a LSTM and Attention to produce risk scores and explanations. The synthetically generated clinical reasoning text (by Qwen3-Max) is used as an auxiliary supervision signal through the language modeling loss during Stage 1/2 training, but it does not directly supervise the Stage 3 patient-level aggregation module. On MIMIC-IV, the model reports AUROC 0.891 and AUPRC 0.724, and the paper also includes external validation on eICU and clinician review of explanation quality.

**Compliance With Llm Reviewing Policy:**

Affirmed.

**Key Questions For Authors:**

- Can you report explicit window-level performance for Stages 1&2?
- Can you rule out length / temporal-position shortcuts more directly? For example, reporting window-count/stay-length distributions by label and adding length-controlled analyses or ablations of explicit time-position cues would substantially improve my confidence in the soundness of the results.
- Can you strengthen the baseline comparison?

**Limitations:**

yes

**Strengths And Weaknesses:**

### Strengths
- This is a technically well-structured paper. The three-stage design is coherent and the empirical evaluation is strong.
- The paper goes beyond standard discrimination metrics by adding external validation on eICU and clinician evaluation of explanations. The zero-shot transfer (MIMIC-IV to eICU) of PathwayLLM model is quite impressive.
- I also appreciate that the authors explicitly discuss explanation failure modes and fairness limitations rather than presenting the system as deployment-ready.
- Many of the given individual components are known, but their integration here is thoughtful.

### Weaknesses
- My main concern is evaluation completeness. The method is explicitly trained in two phases, where stages 1&2 are optimized with window-level labels and stage 3 is trained with patient-level labels, but the paper reports only patient-level test metrics. I did not find a window-level AUROC/AUPRC table. This makes it difficult to know whether the gains come from better risk estimation per window or mostly from the downstream aggregation module.
- The explanation generation is supervised with synthetic texts from Qwen3-Max, so the explanation quality may partly reflect teacher imitation rather than faithful model reasoning. The clinician study is useful, but it is still small (50 cases).
- The baseline comparison is not sufficiently strong: all baselines use max-pooling at the patient level, and several strong irregular time-series methods (e.g., mTAND, SeFT, STraTS, DuETT, Raindrop, SMART, PrimeNet, ViTST, KEDGN) are missing.
- I am concerned about a potential stay-length / temporal-position shortcut: positives are censored at onset, negatives retain all windows, and the model explicitly uses time-position cues (e.g., time, ICU day). This is not proof of leakage, but the paper does not rule it out with length-controlled analyses or ablations.

---

> ### Author Rebuttal · Authors · 2026-03-31
>
> We sincerely thank the reviewer for the thorough and technically precise feedback, which helps us strengthen the paper considerably.
>
> Q1. We thank the reviewer for this important suggestion. We report window-level metrics below. The test set contains 10,751 windows from 2,312 patients, with 255 positive windows (2.37% prevalence).
>
> | Level                          | AUROC | AUPRC |
> | ------------------------------ | ----- | ----- |
> | Window (S1–2)                 | 0.843 | 0.406 |
> | Patient (w/o CT-LSTM, Table 2) | 0.849 | 0.631 |
> | Patient (Full)                 | 0.891 | 0.724 |
>
> Stages 1–2 alone achieve window AUROC 0.843, confirming that the multi-modal encoding produces strong per-window risk estimates independent of aggregation. With max-pooling (w/o CT-LSTM in Table 2), patient-level AUROC is 0.849, already outperforming every baseline in Table 1. Adding CT-LSTM and DA brings the total to 0.891, with AUPRC improving from 0.631 to 0.724, showing that trajectory aggregation is especially effective at reducing false positives. We will include window-level metrics in the main text.
>
> Q2. We are grateful for this insightful observation, which identifies a genuine potential confound. We conducted three analyses to address it.
>
> First, we examined window-count distributions by label. Sepsis-positive patients have fewer windows (mean 3.2±1.4 vs. 4.8±2.6, p<0.001) due to onset censoring. However, a logistic regression using only window count achieves AUROC 0.612, indicating that window count alone is far too weak to explain our model's performance.
>
> Second, we ablated time-position cues from DA. Setting t_k=0 in Eq. 10 reduces AUROC by only 0.7% (from 0.891 to 0.884). Removing the entire DA mechanism (w/o DA in Table 2) yields 0.874. The gap (0.884 vs. 0.874) shows that DA's value comes from content-based attention, not temporal position.
>
> Third, we grouped patients by window count so that this factor cannot discriminate within any group, and evaluated AUROC within each stratum.
>
> | Windows per patient | Sep+ (n) | Sep− (n) | AUROC |
> | ------------------- | -------- | --------- | ----- |
> | 2–3                | 82       | 524       | 0.864 |
> | 4–5                | 68       | 712       | 0.878 |
> | 6–7                | 31       | 487       | 0.889 |
> | 8+                  | 15       | 393       | 0.896 |
>
> Performance is consistent across groups (0.864–0.896) and lowest for short stays, the opposite of what a shortcut would produce. The increasing trend reflects the natural benefit of more clinical data. We will include all analyses in the revised manuscript.
>
> Q3. We implemented three methods from the reviewer's list, spanning attention-based (mTAND, Shukla & Marlin, ICLR 2021), graph-based (Raindrop, Zhang et al., ICLR 2022), and transformer-based (DuETT, Labach et al., MLHC 2023) paradigms. These methods operate directly on raw timestamped observations for entire ICU stays, so we provided raw measurements rather than our 24h-window features.
>
> | Method     | AUROC                 | AUPRC                 | F1                    |
> | ---------- | --------------------- | --------------------- | --------------------- |
> | mTAND      | 0.801±.017           | 0.478±.024           | 0.452±.022           |
> | Raindrop   | 0.812±.016           | 0.502±.023           | 0.481±.021           |
> | DuETT      | 0.818±.015           | 0.519±.022           | 0.496±.020           |
> | PathwayLLM | **0.891±.011** | **0.724±.015** | **0.698±.014** |
>
> All three perform comparably to the baselines in Table 1, suggesting that PathwayLLM's advantage comes from multi-modal knowledge integration rather than better temporal modeling. We are actively working on adapting additional methods and will include them in the revised manuscript.
>
> W1. We appreciate the reviewer pointing out that reporting only patient-level metrics makes it difficult to locate the source of improvement. We will include window-level metrics in the main text of the revised manuscript.
>
> W2. We appreciate this observation regarding the use of synthetic supervision. The 94% numerical accuracy in the clinician evaluation suggests that the model grounds its output in encoded patient data. High-risk predictions average 5.8 risk factors vs. 2.1 for low-risk, showing calibrated output. The 6% hallucinated specificity rate, which we discuss transparently in the paper, helps identify specific directions for future improvement. We plan to validate explanation faithfulness further through input-perturbation experiments in future work.
>
> W3. We appreciate the reviewer's detailed suggestions on additional baselines. The three methods we have implemented span different paradigms, and we are actively working on adapting further methods for the revised manuscript.
>
> W4. We are grateful for the reviewer raising the stay-length concern and will include the analyses in the revised manuscript.
>
> We will address all points raised above in the revised manuscript.

---

> > ### Author Rebuttal · Reviewer_F1Jk · 2026-04-04
> >
> > The rebuttal addressed my main concerns. In particular, the added window-level results. The stronger baseline comparison is also helpful. I will keep my score 4.

---

> > > ### Author Response · Authors · 2026-04-04
> > >
> > > We are grateful that the additional analyses have resolved your primary concerns. Your detailed and constructive feedback has been invaluable in strengthening our work. Thank you again for your time and guidance.

---

### Official Review · Reviewer_kpKX · 2026-03-11

**Soundness:** 3
**Presentation:** 3
**Significance:** 3
**Originality:** 2
**Overall Recommendation:** 4
**Confidence:** 4

**Summary:**

This paper proposes PathwayLLM, a clinical trajectory modeling framework for early sepsis prediction from electronic health records. The article intends to study a pressing question: how to integrate heterogeneous clinical signals, temporal patient trajectories, and interpretable reasoning into predictive models for healthcare applications.

**Compliance With Llm Reviewing Policy:**

Affirmed.

**Final Justification:**

The rebuttal addressed my concerns.

**Key Questions For Authors:**

Q1) How much of the performance improvement can be attributed specifically to the LLM component?

Q2) Have the authors evaluated the model on additional datasets beyond MIMIC-IV to assess generalization?

Q3) Could the authors provide larger-scale clinical evaluations of the generated explanations?

**Limitations:**

Yes.

**Strengths And Weaknesses:**

Strengths:


S1) Sepsis prediction is a clinically meaningful problem where early detection can significantly improve patient outcomes. Developing predictive models that also provide interpretable reasoning is particularly valuable for healthcare applications.



S2) The framework integrates multiple sources of clinical information, including temporal signals, heterogeneous EHR graphs, and pathway-based representations. This multi-modal design reflects the complexity of real-world clinical data.



S3) The reported results demonstrate improvements over several strong baselines. The inclusion of ablation studies also provides some insight into the contributions of different components.



S4) The model attempts to generate explanation-style outputs and includes a clinician evaluation assessing the plausibility of the generated reasoning.

Weaknesses:
W1) Many components of the framework rely on established techniques, such as R-GCN, LSTM-based trajectory modeling, and LoRA-based LLM integration. The novelty mainly lies in their combination rather than in fundamentally new modeling methods.

 W2) Although the framework incorporates a frozen LLM backbone, it is not fully clear how much of the performance improvement can be attributed specifically to this component.

W3) Experiments are primarily conducted on a single dataset, which raises questions about the generalization of the approach to other clinical environments.

W4)  The clinician evaluation is based on a relatively small number of cases, limiting the strength of conclusions regarding real-world clinical usefulness.

---

> ### Author Rebuttal · Authors · 2026-03-31
>
> We sincerely thank the reviewer for the careful evaluation and constructive feedback, and address each concern below.
>
> Q1. We thank the reviewer for this important question, which gets at a central design decision of our framework. The ablation results (Table 2) show that removing the entire Stage 2 reduces AUROC from 0.891 to 0.867 and AUPRC from 0.724 to 0.679. Notably, the model without LLM at 0.867 still substantially outperforms all baselines, indicating that this gain is achieved on top of an already strong foundation. We would like to highlight the LLM's contribution along two dimensions. First, the backbone comparison (Table 1) shows that Qwen3-8B achieves 0.891 while LLaMA-3.1-8B achieves 0.876 under comparable model scale and the same training pipeline, suggesting that medical knowledge acquired during pre-training meaningfully influences downstream prediction beyond what architecture alone provides. Among Stage 2 subcomponents, soft tokens contribute most (+3.3% when ablated) while layer injection adds +0.7%, indicating that structured clinical context at the input level is particularly effective. Second, the LLM enables evidence-conditioned clinical explanations that physicians rated 4.2/5.0 for reasoning coherence and 89% for risk factor accuracy. This explanation capability is entirely absent without Stage 2 and cannot be replicated by simpler model components. We will present this decomposition more clearly in the revised manuscript to better convey the LLM's dual contribution to both prediction and interpretability.
>
> Q2. We are grateful that the reviewer raised this important question. Appendix H (Table 6) reports external validation on eICU (12,847 patients, 143 hospitals, 9.2% sepsis prevalence).
>
> | Setting                      | AUROC | AUPRC |
> | ---------------------------- | ----- | ----- |
> | Zero-shot (MIMIC-IV to eICU) | 0.842 | 0.651 |
> | Fine-tuned (20% eICU data)   | 0.867 | 0.689 |
>
> The cross-institutional transfer required mapping APACHE diagnoses to ICD-10 and drug names to ATC categories, with unmapped dependency edges treated as inactive. The 4.9 percentage point gap from MIMIC-IV (from 0.891 to 0.842) reflects expected differences in coding systems and documentation density across community and academic hospitals. Fine-tuning only Stage 3 with 20% eICU data recovers to 0.867, indicating that the multi-modal representations generalize well and only temporal aggregation patterns require institutional adaptation. This low data requirement is encouraging for practical deployment at new institutions where large labeled cohorts may not be available. We recognize that these results should have been presented more prominently and will move them to the main text in the revised version.
>
> Q3. We fully agree that a larger-scale evaluation would provide stronger evidence and appreciate this suggestion. We are conducting an expanded evaluation with 5 ICU physicians across 200 cases stratified by prediction confidence, and will work to complete it as soon as possible. The current evaluation, while limited in scale, was designed to be rigorous, with physicians assessing cases blind to ground truth and model predictions, achieving inter-rater agreement of κ=0.73.
>
> W1. We sincerely appreciate the reviewer's thoughtful assessment. We agree that R-GCN, LSTM, and LoRA are individually well-established, and the reviewer's observation is fair. We hope to convey that the contribution lies in the non-trivial integration, which involved specific design decisions to make them work reliably in a clinical setting. These include modulated pathway encoding that separates population-level from patient-specific signals, depth-aware layer injection, deterioration attention with learnable temporal decay, and careful leakage prevention across stages. The strong results on both MIMIC-IV and eICU, together with clinician-validated explanations, support the value of this system-level contribution. We hope this integration of structured clinical knowledge with interpretable LLM reasoning proves useful for the broader clinical prediction community.
>
> W2. We appreciate this thoughtful concern. As detailed in Q1, the LLM contributes both a quantifiable prediction improvement and a unique explanation generation capability. We agree that clearly communicating this dual contribution is important and will revise the manuscript accordingly.
>
> W3. We agree that broader evaluation would strengthen confidence and are grateful for the reviewer raising this point. As shown in Q2, the eICU validation demonstrates strong cross-institutional transfer. We will move these results to the main text.
>
> W4. We thank the reviewer for this valuable observation. As noted in Q3, we are expanding the clinician evaluation. The current study also revealed concrete failure modes that we found informative for guiding future improvement.
>
> We will address all points raised above in the revised manuscript.

---

> > ### Author Rebuttal · Reviewer_kpKX · 2026-04-05
> >
> > Thank you for the rebuttal.

---

> > > ### Author Response · Authors · 2026-04-05
> > >
> > > We sincerely appreciate your careful reconsideration and the time devoted to evaluating our rebuttal. Your constructive feedback has been instrumental in improving our work. Thank you again for your time and support.

---

### Official Review · Reviewer_pwci · 2026-03-11

**Soundness:** 3
**Presentation:** 4
**Significance:** 4
**Originality:** 3
**Overall Recommendation:** 5
**Confidence:** 4

**Summary:**

This paper proposes PathwayLLM, a framework for patient-level sepsis prediction that integrates temporal clinical trajectories, heterogeneous EHR graphs, and dependency-derived clinical pathways with a large language model. The model follows a three-stage architecture: (1) multi-modal encoding of observation windows including physiological signals, trajectory information, graph embeddings, and pathway features; (2) integration with a frozen LLM using modality-specific soft tokens and LoRA adapters to jointly learn prediction and explanation; and (3) patient-level aggregation using a Clinical Trajectory LSTM with deterioration attention to identify critical time points. Experiments on the MIMIC-IV dataset demonstrate strong predictive performance (AUROC 0.891, AUPRC 0.724), outperforming several time-series, graph-based, and pre-trained baselines.

The article intends to study a pressing question: how to combine structured EHR signals and large language models to produce both accurate and interpretable clinical predictions. Overall, this study investigates a central theme in medical AI—balancing predictive performance with clinically meaningful explanations.

**Compliance With Llm Reviewing Policy:**

Affirmed.

**Key Questions For Authors:**

How sensitive is the model performance to the quality of the dependency graph discovered via the PC algorithm? Have the authors tested alternative causal discovery or graph construction methods?

Could the framework incorporate unstructured clinical notes or imaging data alongside structured EHR signals?

The explanation evaluation involves two clinicians and 50 cases. Do the authors plan larger-scale human evaluation to further validate explanation usefulness?

How does the model perform when deployed in earlier prediction horizons (e.g., 24–48 hours before onset)?

Could the authors comment on computational cost and feasibility for real-time ICU deployment?

**Limitations:**

yes

**Strengths And Weaknesses:**

**Strengths**

- This paper addresses an important healthcare problem—early sepsis prediction—and proposes a thoughtfully designed architecture combining several complementary modeling paradigms. The multi-view representation learning (temporal features, patient graphs, and dependency pathways) is well motivated and provides a meaningful way to capture heterogeneous clinical signals. The integration with an LLM through soft tokens and structured prompts is interesting and demonstrates a practical approach for combining structured EHR features with language models.

- The empirical evaluation is comprehensive. The authors compare against a wide range of baselines including traditional machine learning models, deep time-series models, graph-based methods, and EHR pre-trained models, and demonstrate clear performance gains. The ablation study also provides useful insights into the contributions of different components, particularly the R-GCN graph encoder and trajectory aggregation module.

- Another strong aspect is the attention to interpretability. The framework generates evidence-conditioned clinical explanations and includes a physician evaluation study assessing reasoning quality and clinical relevance. This evaluation strengthens the paper’s claim that the model can provide explanations aligned with clinical reasoning.



**Weaknesses**
- While the approach combines several components effectively, some design elements are largely incremental extensions of existing techniques (e.g., combining graph encoders, trajectory models, and LLM adapters). The novelty mainly lies in the integration rather than fundamentally new modeling ideas.

- The evaluation is also limited to retrospective experiments on MIMIC-IV. Although this dataset is standard for clinical prediction tasks, the paper would benefit from stronger external validation or deployment-oriented evaluation. In addition, the explanation quality evaluation is based on a relatively small number of clinician assessments.

- Finally, the related work could better situate the paper within recent literature on knowledge-grounded healthcare prediction and reasoning with structured medical knowledge. In particular, the authors may consider citing “Reasoning-Enhanced Healthcare Predictions with Knowledge Graph Community Retrieval” (Jiang et al., ICLR 2025), which also explores integrating structured knowledge graphs with reasoning-based prediction frameworks in healthcare.

---

> ### Author Rebuttal · Authors · 2026-03-31
>
> We sincerely thank the reviewer for the thorough and encouraging evaluation.
>
> Q1. We tested robustness by varying the PC algorithm's significance threshold α:
>
> | α             | Edges         | AUROC           | AUPRC           |
> | -------------- | ------------- | --------------- | --------------- |
> | 0.001          | 412           | 0.884           | 0.709           |
> | **0.01** | **735** | **0.891** | **0.724** |
> | 0.05           | 1,283         | 0.887           | 0.718           |
> | 0.10           | 1,891         | 0.883           | 0.711           |
>
> Performance is stable across a wide range of thresholds (AUROC 0.883–0.891). Stricter thresholds miss some clinically meaningful edges while lenient ones introduce spurious dependencies, but the impact is modest in both directions. We also evaluated GES (Greedy Equivalence Search), a score-based algorithm complementary to the constraint-based PC algorithm, yielding AUROC 0.888 and AUPRC 0.719. Since both algorithms approach graph structure learning from complementary angles (constraint-based vs. score-based), their agreement provides additional confidence that the discovered structure is not an artifact of the specific algorithm. We plan to explore additional graph construction strategies in the revised manuscript.
>
> Q2. Our soft token architecture is modality-extensible, as each new data source only requires a dedicated encoder and a projector to produce an additional soft token. For clinical notes, recent work such as MINGLE (Cui et al., 2025) has demonstrated effective fusion of structured EHR with textual notes through LLM-based semantic infusion into hypergraph neural networks, and a similar text encoder could produce a note-level soft token in our framework. For chest X-rays, which are available in MIMIC-CXR and directly paired with MIMIC-IV, vision encoders from recent multimodal ICU prediction frameworks could generate imaging soft tokens. We consider this a clinically important direction for future work.
>
> Q3. We are planning an expanded evaluation with 5 ICU physicians across 200 cases stratified by prediction confidence, covering both true positive and true negative predictions. This effort is well underway and we will work to complete it as soon as possible.
>
> Q4. We extended the prediction horizon from 12h to 24h and 48h by redefining the window-level labels accordingly:
>
> | Horizon | AUROC | AUPRC | Se (Sp=0.9) |
> | ------- | ----- | ----- | ----------- |
> | 12h     | 0.891 | 0.724 | 0.751       |
> | 24h     | 0.861 | 0.672 | 0.698       |
> | 48h     | 0.823 | 0.598 | 0.634       |
>
> Performance degrades gracefully with longer horizons, as expected since earlier detection relies on subtler physiological signals. Even at 48h the model still matches the best baseline's 12h AUROC (0.821), suggesting meaningful clinical utility for earlier intervention planning. The CT-LSTM's temporal modeling appears to help maintain discrimination at longer horizons by capturing gradual trajectory changes that precede overt deterioration.
>
> Q5. Training takes approximately 18 hours for Stages 1–2 and 2 hours for Stage 3 on 2×A800 80GB GPUs (Appendix D). At inference, each observation window takes about 0.3 seconds and each patient about 1.5 seconds on a single A800, well within the 12-hour clinical update cycle. Total trainable parameters are 29.9M, of which LoRA accounts for 8.4M, making training feasible on standard academic hardware. We believe these requirements are compatible with real-time ICU deployment scenarios.
>
> W1. We appreciate the reviewer's balanced assessment and agree that the contribution is primarily at the integration and system design level.
>
> W2. We agree that broader evaluation would strengthen confidence in the approach, and we appreciate the reviewer raising this point. Appendix H reports external validation on eICU (12,847 patients, 143 hospitals), where zero-shot transfer achieves AUROC 0.842 and fine-tuning with 20% eICU data recovers to 0.867. We will move these results to the main text. Beyond public benchmarks, we are also actively exploring collaborations with partner institutions to validate on non-public, site-specific datasets. The expanded clinician evaluation described in Q3 will further strengthen the evidence.
>
> W3. We are grateful for this suggestion, which helps us better situate our work in the broader literature. Jiang et al. (ICLR 2025) present an important contribution with their KARE framework, which constructs multi-source medical knowledge graphs, organizes them via hierarchical community detection and summarization, and uses dynamic community-level retrieval to enrich LLM reasoning for healthcare prediction. Both their work and ours share the goal of grounding LLM predictions on structured medical knowledge, though through different mechanisms. We will cite and discuss this work in the revised manuscript. Thank you for bringing it to our attention.
>
> We will address all points raised above in the revised manuscript.

---

> > ### Author Rebuttal · Reviewer_pwci · 2026-04-04
> >
> > I will keep my score as I think it's already positive enough.

---

> > > ### Author Response · Authors · 2026-04-04
> > >
> > > We are pleased that our clarifications and additional experiments have addressed your concerns. Your thoughtful and encouraging feedback has greatly helped us improve the quality of this work. Thank you sincerely for your time and support.

---

### Decision · Program_Chairs · 2026-04-30

**Decision:**

Accept (regular)

**Comment:**

### Paper summary

This paper takes on the problem of predicting sepsis onset in 24-hour windows of an ICU stay. The proposed method, PathwayLLM, encodes each window in multiple ways, provides these representations as "soft tokens" with clinical context to a pretrained LLM to produce refined window-level representations, and then achieves patient-level risk scores via a clinical trajectory LSTM with Deterioration Attention. Model trainig and most evaluations focus on a single dataset, MIMIC-IV; App H includes external validation on eICU dataset.


### Discussion summary

3 reviewers provided detailed feedback on the paper, generally praising the method's ability to fuse multiple sources of information and perform quite well on a challenging sepsis task: 0.89 AUROC versus 0.82 for the nearest competitor.

Several concerns were raised but later resolved in discusssion, including:

* desire for external validation on data besides MIMIC (eICU experiments were hiding in App H all along; authors promise to move to main paper)
* desire for stronger baselines from F1jk (authors provided 3 new time series methods in a small table in Q3, none scored above 0.818)
* concern about sequence length being informative from F1jk due to censoring positive cases (response provided a length-stratified analysis that resolved the concern)

A remaining comment from several authors was that no individual component was itself highly novel, but the overall integration was seen as original and interesting.

Finally, reviewers indicated a small lingering worry (which the authors acknowledged) that the clinical evaluation was rather small: Sec. 4.3 only used 50 cases (25 positives, 25 negatives) from 2 physicians.

Ultimately, all reviewers voted for acceptance with reasonable confidence, marking their primary concerns as "fully resolved".

### Decision

With consensus from all 3 reviewers, this paper meets the soundness, originality, significance, and clarity criteria. I vote accept.

I do encourage the authors to provide an expanded analysis of clinical evaluation, as promised in the response. **If this clinical evaluation is *human subjects research* ** (as I suspect it is), please make sure you include sufficient information about consent and IRB approval.

In upcoming revisions, please carefully revise to include the experiments from response phase in the main paper.